# Evaluation of Complexity Measurement Tools for Correlations with Health-Related Outcomes, Health Care Costs and Impacts on Healthcare Providers: A Scoping Review

**DOI:** 10.3390/ijerph192316113

**Published:** 2022-12-01

**Authors:** Hiromitsu Kaneko, Akiko Hanamoto, Sachiko Yamamoto-Kataoka, Yuki Kataoka, Takuya Aoki, Kokoro Shirai, Hiroyasu Iso

**Affiliations:** 1Faculty of Medicine, Osaka University, Suita, Osaka 565-0871, Japan; 2Independent Researcher, Kita-ku, Kyoto 603-8233, Japan; 3Department of Health Informatics, Kyoto University Graduate School of Medicine/School of Public Health, Yoshida Konoe-cho, Sakyo-ku, Kyoto 606-8501, Japan; 4Department of Internal Medicine, Kyoto Min-Iren Asukai Hospital, Tanaka Asukai-cho 89, Kyoto 606-8226, Japan; 5Scientific Research Works Peer Support Group (SRWS-PSG), Osaka 541-0043, Japan; 6Section of Clinical Epidemiology, Department of Community Medicine, Kyoto University Graduate School of Medicine, Shogoin Kawara-cho 54, Kyoto 606-8507, Japan; 7Department of Healthcare Epidemiology, Kyoto University Graduate School of Medicine/School of Public Health, Yoshida Konoe-cho, Kyoto 606-8501, Japan; 8Division of Clinical Epidemiology, Research Center for Medical Sciences, The Jikei University School of Medicine, 3-25-8 Nishishimbashi, Minato-ku, Tokyo 105-8461, Japan; 9Department of Social Medicine, Osaka University Graduate School of Medicine, Osaka 565-0871, Japan; 10Institute for Global Health Policy Research, Bureau of International Health Cooperation, National Center for Global Health and Medicine, Tokyo 162-8655, Japan

**Keywords:** patient complexity, scoping review, tools, primary care, health-related outcomes

## Abstract

Various tools to measure patient complexity have been developed. Primary care physicians often deal with patient complexity. However, their usefulness in primary care settings is unclear. This study explored complexity measurement tools in general adult and patient populations to investigate the correlations between patient complexity and outcomes, including health-related patient outcomes, healthcare costs, and impacts on healthcare providers. We used a five-stage scoping review framework, searching MEDLINE and CINAHL, including reference lists of identified studies. A total of 21 patient complexity management tools were found. Twenty-five studies examined the correlation between patient complexity and health-related patient outcomes, two examined healthcare costs, and one assessed impacts on healthcare providers. No studies have considered sharing information or action plans with multidisciplinary teams while measuring outcomes for complex patients. Of the tools, eleven used face-to-face interviews, seven extracted data from medical records, and three used self-assessments. The evidence of correlations between patient complexity and outcomes was insufficient for clinical implementation. Self-assessment tools might be convenient for conducting further studies. A multidisciplinary approach is essential to develop effective intervention protocols. Further research is required to determine these correlations in primary care settings.

## 1. Introduction

The concept of health includes mental and social well-being as well as physical well-being [1]. The conventional biomedical model requires that disease be treated as an entity independent of social behaviors; however, it is currently widely recognized that disease is a multifaceted concept that is not fully captured by the number or type of medical conditions or by previous healthcare costs [2,3]. To overcome the limitations of the conventional biomedical model, the concept of “patient complexity” has been proposed.

To date, there is no established definition of patient complexity. Instead, several explanations with different scopes have been suggested [4,5,6,7,8,9,10]. According to Nicolaus [11], patient complexity is a dynamic state in which biomedical, mental health, socioeconomic factors, and patient preferences are affected by each other. To deal with the patient complexity, conceptual models and complexity measurement tools are recognized. Some studies have developed conceptual models that describe complexity as the relationships among domains, such as diagnostic procedures, treatment strategies, interventions, multiple healthcare providers, or patients’ behaviors [4,5,9,10]. Conceptual models can suggest holistic approaches to address broader issues but sometimes are impractical for clinical implementation.

In contrast, other studies have developed tools to make complexity more easily quantifiable [6,7,8]. For instance, INTERMED evaluates biological, psychological, social, and healthcare domains through interviews in which scores are assigned to 20 items on a scale of 0 (no symptoms) to 3 (severe symptoms). Tools can only be applied to specific situations for which they are designed, such as for adult inpatients, but are more applicable to clinical cases. Various tools for measuring patient complexity have been developed for specific purposes and settings.

In this scoping review, we focused on both observational and interventional studies that examined outcomes related to complex patients stratified by complexity measurement tools. Contrary to our review article, previous studies reviewed these complexity measurement tools focusing on particular diseases [12]. While there exists a versatile range of tools, previous reviews did not clearly elucidate what kinds of measures could be used with the general adult populations or with patients but not in the context of specific diseases or settings, such as the ICU (intensive care unit) and home healthcare. Moreover, previous reviews did not investigate how complexity measurement tools are related to outcomes of general adult populations or patients, and the performance of healthcare providers/systems. Some studies described the characteristics of adult populations that are frequent users of healthcare providers/systems [13]; however, they did not examine the correlation with patient complexities.

Therefore, this scoping review clarified the general status of tools to measure patient complexity for the general adult population or patients, the association between patient complexity measured by these tools and patient outcomes, and the associated impact on healthcare providers/systems, especially from the perspective of primary care physicians. While specialists are concerned essentially with identifying whether a patient has the disease in which the specialist is an expert, primary care physicians as generalists must recognize a broad range of problems [14]; therefore, the care of highly complex patients is one of the specialties of primary care physicians, and such complexity is an essential concept in primary care. Accordingly, we focus particularly on the perspectives of primary care physicians.

## 2. Materials and Methods

As we described in the research protocol logged at the OSF (https://osf.io), we conducted this review following the PRISMA extension for scoping reviews (PRISMA-ScR) statement [15]. We also used the scoping review framework of the Joanna Briggs Institute (JBI) [16], following the five-stage approach listed below.

### 2.1. Stage 1: Identifying the Research Question

We were interested in the patient complexity of the general population or patients. The research questions were as follows:

RQ1. What tools to assess patient complexity are available for the general populations or patients?

RQ2. To what extent do health related patient outcomes differ according to the magnitude of the measured indicators?

RQ3. To what extent does the impact on healthcare providers/systems (e.g., healthcare costs, administrative costs, medical staff burnout) differ depending on the magnitude of the measured indicators?

### 2.2. Stage 2: Identifying Relevant Studies

We used the population, concept, and context (PCC) framework [16] to define the inclusion criteria as follows:

#### 2.2.1. Participants

We included studies that considered adult general populations or patients (18 years and older). Studies were excluded if they were limited to specific conditions rather than general adult populations or patients, such as psychiatric patients, pediatric patients, and pregnant women. We excluded studies that focused only on patients with specific diseases, such as diabetes, heart failure, and rheumatoid arthritis. In addition, studies that included patients in particular settings (e.g., postoperative, ICU, home healthcare) were excluded. However, inpatients were included if this was not in a disease-specific context.

#### 2.2.2. Concept

We reviewed extant literature that evaluated patient outcomes (e.g., health outcomes, disease control, patient quality of life) and the impact on healthcare providers/systems (e.g., healthcare costs, administrative costs, healthcare professional burnout) through complexity measurement tools. In order to cover related studies broader, we included not only observational studies but also interventional studies that could examine correlations between patient complexity stratified by tools and outcomes at baseline.

We defined patient complexity as “a typology of 4 overarching categories contributing to patient complexity: medical complexity, mental health disorders, socioeconomic factors, and individual patient behaviors or traits” [6]. Therefore, we excluded some tools which mainly focused on biomedical aspects such as comorbidity/multimorbidity measurement tools, end-of-life/palliative specific measures. In addition, we intended to examine the correlations between patient complexity as “exposure” and measured outcomes. We excluded tools which focused on predicting frequent visit/hospitalization, and panel-weighting management tools machine learning prediction tools/algorithms, and software tools, because they focused on predicting outcomes rather than measuring them or examining correlation with patient complexities. For instance, we did not include the frailty score [17], which is widely used and embedded in the electronic patient record systems in the United Kingdom.

We included criterion-related validity studies that validated the performance of the measurement tools against some other criteria, such as discriminative and predictive validities. However, studies of concurrent validity that analyzed correlations between other complexity measurement tools were excluded. All reliability studies other than those that addressed criterion-related validity, such as agreement in measurement results among examiners, were excluded because these reliability studies did not contain outcomes such as patient health related outcomes, healthcare costs, or impacts on healthcare providers.

#### 2.2.3. Context

In this review article, as we stated in the previous section, we defined patient complexity as “a typology of 4 overarching categories contributing to patient complexity: medical complexity, mental health disorders, socioeconomic factors, and individual patient behaviors or traits” [6]. Followed by this definition, we analyzed the title, abstract, and index terms used to characterize the articles found in the preliminary search. We paid attention to select search terms so that they included “patient complexity” and related expressions, and tools or indicators which measured complexity. We also included studies of “patient complexity” as defined by the authors in any setting (including primary and secondary care settings) from any country. We searched the data for all available lengths of each database up until July 2022 without limiting the starting date of the search period. We searched two databases MEDLINE and CINAHL (see Appendix A for search terms and search string). We also checked references from prior systematic reviews [11,12,13], and searched for references in the articles identified for inclusion, using Web of Science. If a study was not indexed in Web of Science (http://webofscience.help.clarivate.com/) (Accessed on 30 June 2022), we used Citationchaser [18] or Google scholar (https://scholar.google.com/intl/en/scholar/about.html) (Accessed on 30 June 2022). All published observational studies with controls, case reports (more than 10 reported cases), and case series were included. Studies in English and from any country were accepted, and we included studies with any length of follow-up. We excluded conference abstracts and review articles. In addition, we consulted a GP in London, an expert of patient complexity management. He gave us some advice without additional papers.

### 2.3. Stage 3: Study Selection

We selected studies following the Preferred Reporting Items for Systematic Reviews and Meta-analysis (PRISMA) flow diagram. We used the PCC framework. Two of three reviewers (HK, AH and SYK) selected the abstracts independently. One reviewer (HK) selected the full texts and another (AH or SYK) confirmed the decision. Any differences in opinion were resolved by discussion and, if resolution was not possible, through arbitration by a third researcher (YK). We also conducted a citation search for included articles using several search data bases, as we stated in the previous section.

### 2.4. Stage 4: Charting the Data

Data extraction was carried out by one researcher (HK) using standard data extraction forms, including the names of authors, year, and country in which each screening tool was developed. HK also extracted the population screened by the tool, the patient outcome (e.g., probability of a person being hospitalized within one year), impact on healthcare providers/systems, and the costs. The other two researchers (AH and SYK) confirmed the data extraction. If there were disagreements, they were resolved through discussion and extraction of related data.

### 2.5. Stage 5: Collating, Summarizing and Reporting the Results

We extracted the complexity measurement tools and the studies that examined patient outcomes, healthcare costs, and impacts on healthcare providers using complexity measurement tools. For each complexity management tool, we counted and listed the number of studies that examined patient outcomes, healthcare costs, and impacts on healthcare providers. Patient outcomes were classified into 19 detailed categories, as shown in Appendix A. We listed categories measured in correlation with complexity in the outcome measurement column of Table 1. If a significant correlation was not observed, related categories on the list were underlined. While the intervention studies that implicitly investigated correlations between patient complexity and outcomes were included, we counted them as no correlations with complexity.

### 2.6. Differences between Protocol and Review

We expanded the scope of the participants from the general adult population to the general adult population or patients because we recognized that studies related purely to the former, namely community dwelling citizens, were few in number. We defined general adult patients as those not with a specific disease and not in specific settings, such as the ICU or home healthcare. We reduced and clearly defined the complexity measurement tools included in this review by excluding tools such as comorbidity/multimorbidity measurement tools, frequent visit/hospitalization prediction tools, end-of-life/palliative specific measures, general condition panel-weighting management tools, machine learning prediction tools/algorithms, and software tools.

## 3. Results

### 3.1. Patient Complexity Measurement Tools Used with the General Adult Population or Patients

After removing duplicates, we identified 933 records via databases. After title, abstract, and full text review, eight studies were identified, which referred to 17 complexity measurement tools. We identified 18 original tool development studies related to the extracted 17 complexity measurement tools (Table 1). If there were unidentified complexity measurement tools, we searched the tool development studies. Finally, we identified 21 complexity management tools and the corresponding 22 tool development studies (Table 2). Nine out of the 21 tools were used predominately in primary care settings, such as with outpatients or community dwelling people. The other 12 tools were used only for inpatients. Regarding the format of the tools, 11 were designed for face-to-face interviews, 7 were intended to extract data from medical records, and the other 3 were self-assessments.

### 3.2. Correlations between Patient Complexity and Patients’ Health Related Outcomes

In the next step, we conducted a citation search of the 22 tool development studies. We identified 290 studies. Among them, we identified 25 studies (eight tools) that evaluated correlations between patient complexity and outcomes (including criteria-related validity; Figure 1, Table 1). Six out of 25 outcome studies were interventional [21,27,31,35,38,41] (See Appendix A for excluded studies during the process of full-text review). While 8 out of 22 complexity measurement tools were used to stratify participants by complexity profiles to measure correlations between complexity and outcomes, the other 14 tools have not yet been used for that purpose (Table 3). We identified 19 health-related patient outcomes. INTERMED and its derived tools were associated with studies that considered 17 outcomes. The outcome coverage of COMPRI, PCAM, SWAAT, Bandini’s tool, and COMPLEXedex were 4, 3, 3, 2 and 1, respectively. The length of hospital stay was the outcome most examined through complexity measurement tools, which directly referenced patient health condition. Following the length of hospital stay, mental health and health-related quality of life were the variables most frequently examined (Appendix A).

### 3.3. Correlations between Patient Complexity and Impacts on Healthcare Providers/Systems

PCAM is the only measurement tool that was used to describe the burden of health-related staff. IM-E and IM-E-SA were used to evaluate healthcare costs related to complex patients.

## 4. Discussion

This scoping review identified 21 complexity measurement tools. While 9 out of 21 tools were used in primary care settings, the others were used only for inpatients. We identified 25 studies that investigated the correlation between patients’ complexity and health-related outcomes [19,20,21,22,23,24,25,26,27,29,30,31,32,33,34,35,36,37,38,40,41,42,43]. Two studies investigated the correlation between patients’ complexity with healthcare related costs [25,28]. We found only one study that investigated the correlation between patients’ complexity and impacts on healthcare providers [39].

To the best of our knowledge, this is the first scoping review to comprehensively search for patient complexity measurement tools and reveal the correlation between patients’ complexity with outcomes, healthcare costs, and impact on healthcare providers in the general population or patients. We found three previous reviews that investigated complexity measurement tools. Marcoux et al. (2017) focused on the risk of high use of healthcare services associated with complex healthcare needs by using complexity screening tools [13]. They identified 14 different screening tools. We included 4 out of these 14 tools, namely IM, IM-E, IM-SA and IM-E-SA. We excluded the other 10 tools, which were specialized for predicting high frequency of healthcare use or targeted specific populations, such as homeless persons, rather than the general population or patients. Hawker et al. (2021) identified complexity tools that focused on rheumatic disease [12]. They identified two complexity measurement tools for patients with rheumatic disease, namely INTERMED and SLENQ. SLENQ is specialized for patients with systemic lupus erythematosus. Nicolaus et al. (2022) reviewed definitions of patient complexity, which included complexity defined with complexity measurement tools or conceptual models [11]. They concluded that there exists considerable heterogeneity in definition of patient complexity. None of the three review articles focused on tools used in general populations or patients, or outcomes (health-related patient outcomes, healthcare costs, and impacts on healthcare providers).

Theoretically, and especially in primary care settings, the concept of complexity can depict actual clinical situations better than the conventional biomedical model [11]; however, the evidence is insufficient for clinical implementation in that the correlations between patient complexity and outcomes were not clear, and any intervention protocols were not definitely effective for patients stratified by complexity measurement tools. While specialists are concerned essentially with identifying whether the patient has the disease in which they are expert, primary care physicians as generalists must recognize a broad range of problems [14]. This partially explains why the challenges faced by primary care physicians in their workday are not captured sufficiently by conventional biomedical scales. Therefore, complexity measurement tools are required, especially in primary care settings, to determine problems and opportunities [7].

We found that primary care clinicians do not currently have sufficient supportive evidence to routinely measure complexity to improve health-related outcomes based on identified studies. We identified 6 intervention studies and 19 observational studies that examined the correlation between patient complexity and health-related outcomes. In the observational studies, health-related outcomes were generally correlated with complexity (Table 1); however, insufficiently many studies found clear correlations between patient complexity and health-related outcomes. Although the number of studies were limited, we found some studies that showed greater complexity was related to longer duration of hospital stays, worse health-related quality of life, and more mental health problems. Fewer studies focused on various other exposures such as nutrition, substance misuse, and oral health (Appendix A).

Regarding intervention studies, although some theoretical approaches have been attempted [9], there are currently no practical, effective intervention protocols for patients’ complexity. We included six intervention studies [21,27,31,35,38,41] that tracked the time development of outcomes related to complex patients stratified by tools. Four out of six studies [27,31,35,41] did not meaningfully improve health-related outcomes. One found a reduction of the length of hospital stays merely for aged 65 years or older, while no significant effect on that was found for the whole sample [21]. The other one suggested that clinical informational workflows that incorporate social determinants of health data can improve outcomes for patients; however, practical intervention protocol was not proposed [38].

Researchers should conduct additional studies to investigate the correlation between complexity and outcomes in the primary care setting, where the number of complex patients is increasing because of various factors, such as an aging population, increasing prevalence of multimorbidity, and polypharmacy. Researchers should also pay attention to primary care physicians’ situation, in that they usually work understaffed and sometimes must treat a range of complex patients. Some interview tools have been developed specific to the primary care setting [57,58,61]; however, no studies to date have evaluated outcomes. One reason for this may be the time required for measurement.

To develop further research by reducing the burden of measurement, self-administered tools might be preferable than interviewing. For example, one of the interview tools (MCAM) was found to be cumbersome and time-consuming [44]. We included three self-assessment tools in this review, namely IMSA, IM-E-SA and CONECT-6. The IM-E-SA can be utilized in the primary care setting, albeit for older adults [30]. The INTERMED self-administered (IMSA) [49] is another candidate for evaluating outcome, even in primary care settings and not only for older persons. IMSA was developed with reference to IM-E-SA, which was developed and validated four years prior to IMSA. CONECT-6 is used only in emergency department settings. In addition, no study has investigated the correlation with outcomes.

Regarding the area to be investigated in the future, these tools will likely be used to investigate the association between patient complexity and outcomes. In particular, focusing on healthcare costs and impacts on healthcare providers are warranted. To determine effective intervention protocols, a multidisciplinary approach is essential; however, no studies have considered sharing information or action plans with multidisciplinary teams while measuring outcomes for complex patients stratified using tools. Studies of the usefulness of tools to measure complexity would be valuable for information sharing and action planning with multidisciplinary teams.

There are several limitations in our study. First, to develop the search formula in this review, we focused on words related to “complexity.” Because we did not search for constructs related to the concept of complexity, we could have omitted complexity tools that described the concept of complexity but did not include words related to “complexity” per se. Second, only two databases were searched, and only papers in English were selected. This might miss some relevant articles. To overcome this problem, we conducted a citation search with multiple search engines. Third, we conducted a scoping review; therefore, this research does not calculate correlations between patient complexity and outcomes. Fourth, our review did not include perspectives of other health care providers from a primary care perspective. Further review is warranted for other perspectives.

## 5. Conclusions

This scoping review found that 8 out of 21 patient complexity tools were used in studies that evaluated correlations with outcomes. Further research is required to reveal the correlations between patient complexity and outcomes in the primary care setting.

## Figures and Tables

**Figure 1 ijerph-19-16113-f001:**
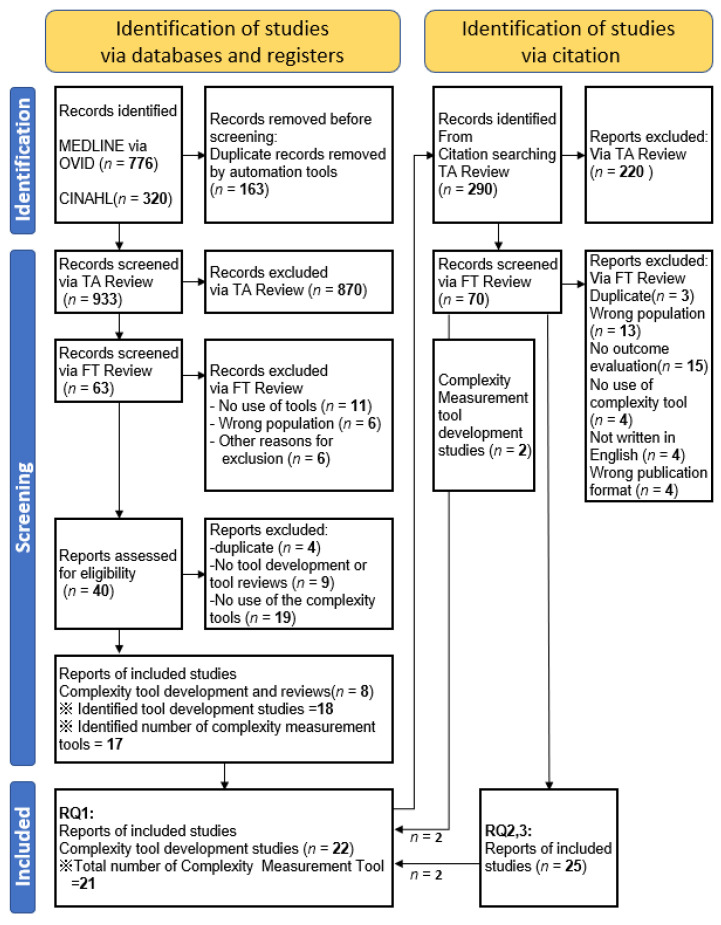
PRISMA flow chart.

**Table 1 ijerph-19-16113-t001:** Characteristics of included studies.

Tool	Author (Year)	Study Design	Setting	*n*	Location	Outcome Measurement
INTERMED	Jonge (2001) [19]	Cohort study	Consecutively admitted patients on a general internal medicine ward	89	Netherlands	(1) Length of hospital stays (2) Number of medications (3) Nurse care intervention (4) Specialist consultation
	Jonge (2003) [20]	Cohort study	Patients from admission to discharge	275	Netherlands	(1) Length of hospital stays (2) Physical health (3) Mental Health
	Jonge (2003) [21]	Interventional study	Inpatients	193	Netherlands	(1) Length of hospital stays (2) Health related quality of life
	Lobo (2015) [22]	Cohort study	Internal medicine inpatients	626	Spain	(1) The number of medical diagnoses (2) The length of hospital stays (3) The cumulative illness rating scale (CIRS) score (4) Number of psychiatric referrals
	C.A. Oliveira [23]	Cohort study	All adults who arrived at the reception of one of the health services	230	Brazil	(1) Quality of life (2) Symptoms of anxiety and depression (3) Social support (4) Comorbidity levels (5) Primary Health Care Use
	H.S. da Silva [24]	Cross sectional and analytical study	Inpatient	382	Brazil	(1) Cognitive decline (2) Activities of daily living
INTERMED for Elderly	B. Wild (2014) [25]	Cohort Study	General health check-up people	3121	Germany	(1) Health-related quality of life (2) Depression symptom severity of generalized anxiety disorder (3) Health care costs
	F.H. Boehlen (2017) [26]	Cohort Study	Health check-up patients	9949	Germany	(1) Personal resources (e.g., Optimism) (2) Social resources (free-time, family, external support)
	R.J. Uittenbroek (2018) [27]	RCT	All adults aged 75 and older, listed with these GP	1456	Netherlands	(1) Health-related quality of life (2) The number of days aging in the place (Robustness)
IM-E-SA	L.L. Peters (2015) [28]	Population-based cohort	General population	713	Netherlands	Healthcare costs
	M.H. Bakker (2020) [29]	Cohort Study	Community-dwelling elderly	89	Netherlands	Oral status and oral health
	A.R. Hoeksema (2017) [30]	Crosssectional descriptive study	Community-dwelling elderly	1026	Netherlands	Oral status and oral health
	R.J. Uittenbroek (2018) [27]	RCT	All adults aged 75 and older, listed with GPs	1456	Netherlands	(1) Health-related quality of life (2) The number of days aging in the place (Robustness)
	S.L. Spoorenberg (2019) [31]	RCT	Community (follow-up)	136	Netherlands	Geriatric ICF (International Classification of Functioning) Core Set (1) Mental health (2) Physical health (3) Nutrition (4) Support/Social Resources
	L.L. Peters (2013) [32]	Cross-sectional study	General population	338	Netherlands	(1) Life satisfaction (2) Activities of daily living (3) Quality of life (4) Mental health (5) Prevalence of diseases/disorders
	Dortland (2017) [33]	Cohort study	Inpatients and outpatients	850	Germany France Italy Netherlands Switzerland	(1) Mental health (2) Medical health (3) Health related quality of life
	M.H. Bakker (2018) [34]	Cross sectional observational study	Community-living elderly	1325	Netherlands	(1) Nutritional status (2) Oral status and health problems
	S.L. Spoorenberg (2018) [35]	RCT	Inpatients and outpatients	850	Germany France Italy Netherlands Switzerland	(1) Health related quality of life (2) Health care use
COMPRI	Jonge (2003) [20]	Cohort study	Patients from admission to discharge	275	Netherlands	(1) Length of hospital stays (2) Physical health (3) Mental Health
	Jonge (2003) [21]	Interventional study	Inpatients	193	Netherlands	(1) Length of hospital stays (2) Health related quality of life
	D. Yokokawa (2022) [36]	Case control study	Newly hospitalised patients	33	Japan	Length of hospital stays
PCAM	S. Yoshida (2017) [37]	Prospective cohort study	Inpatient	201	Japan	Length of hospital stays
	Hewner (2017) [38]	Interventional study	Patients who were discharged to the community	419	USA	(1) Hospitalizations rate (2) Emergency department visit rate (3) Outpatient utilization rate
	S. Yoshida (2018) [39]	Cohort study	All inpatients admitted to the acute care unit of Hospital	201	Japan	Burden for health-related staff
	Y. Sugiyama (2020) [40]	Cross-sectional study	Outpatient practices	426	Japan	Alcohol misuse
COMPLEXedex	S. Hawner (2014) [41]	Before-and-after study	Regional health plan in 2009	411,407	USA	Hospital inpatient utilization
SWAAT	Boutin-Foster (2005) [42]	Cohort	Newly hospitalised patients	299	USA	(1) Length of hospital stays (2) Proportion of patients who were seen by a social worker (3) Proportion of patients who received services at discharge
Bandini’s tool	Bandini (2018) [43]	Observational study	Inpatients	240	Italy	(1) Length of hospital stays (2) Probability of home discharge (3) Risk of death

Note: [] means reference number. Interventional studies are [21,27,31,35,38,41]. Notation: LOS: Length of hospital stays, HLQOL: Health related Quality of life. While the underlined item in an Outcome Measurement column were not observed correlations with complexities, other items in the column shows correlations with complexity scores.

**Table 2 ijerph-19-16113-t002:** Complexity measurement tools, intended use, applied setting, and format.

Complexity Tool	Full Name of Tool	Intended Use of the Tool	Applied Setting	Format of the Tool
IM [44,45]	INTERMED	Indication for multidisciplinary care	Inpatients	Face to face interview
IMSA Dortland	INTERMED Self-Assessment	Self-assessment version of IM	Inpatients	Self-assessment
IM-E [46]	INTERMED for the Elderly	Elderly version of IM	Inpatients, outpatients, community dwelling people	Face to face interview
IM-E-SA [32]	INTERMED for the Elderly Self-Assessment	Self-assessment version of IM-E	Inpatients, outpatients, community dwelling people	Self-assessment
COMPRI [47]	Complexity Prediction Instrument	Indication for multidimensional assessment and interdisciplinary care coordination	Inpatients	Face to face interview
PCAM [48]	Patient Centered Assessment Method	Identify biopsychosocial complexities; make appropriate referrals	Inpatient, Outpatient	Face to face interview
COMPLEXedex [49]		Rank individuals with multiple chronic diseases hierarchically into segments	Inpatients, Community dwelling people	Calculated using the amount of data in the medical records
CONECT-6 [50]	COmplex NEeds Case-finding Tool-6	Identify patients with chronic conditions and complex health needs in emergency Departments	Emergency departments	Self-assessment
IMECSs [51]	Iowa Medication Complexity Scores	Describe challenges a patient faces in developing medication regimen	Outpatients	Calculated automatically in real time using clinical documentation system
CCMR [52]	Cognitive complexity of the medical record	Use CCMR as a useful surrogate for true patient complexity	Inpatients	Calculated using the amount of data in the medical records
Electric order volume [53]		Use electronic order volume a useful surrogate for the workload of residents	Inpatients	Calculated using the data in the medical records
PCA [54]	Patient Complexity Assessment score	Use readily available administrative and clinic data to identify complex inpatients	Inpatients	Calculated using the data in the medical records
PCCL [55]	Patient clinical complexity level	Indicate a calculated index of disease burden per patient based on the amount and constellation of secondary diagnoses	Inpatients	Calculated using the data in the medical records
FADOI-Complimed [56]	Federation of Associations of Hospital Doctors on Internal Medicine Complimed	Describe the complexity of hospitalized patients as a two-dimensional phenomenon	Inpatients	Face to face interview
MCAM [57]	Minnesota Complexity Assessment Method	Identify factors interfering with care; formulate new care plan	Outpatients	Face to face interview
MECAM [58]	Minnesota Edinburgh Complexity Assessment Method	Identify factors posing risk to patient well-being	Outpatients	Face to face interview
OCCAM [59]	Oxford Case Complexity Assessment Method	Identify factors interfering with care; facilitate care coordination	Inpatients	Face to face interview
Corazza’s score [60]		Comprehensive assessment of clinical complexity in which an overall index results from the sum of each vector/domain	Outpatients	Face to face interview
Salisbury’s score [61]		Develop a valid and reliable measure of the complexity of general practice consultations	Outpatients	Face to face interview
SWAAT [42]	Social Work Admission Assessment Tool	SWAAT was developed to identify patients who have complicated discharge planning needs and require early social work evaluation.	Inpatients	Face to face interview
Bandini’s tool [43]		To identify patients with clinically complex hospitalization events	Inpatients	Extracted data from the hospital medical records

Note: [] are the studies which developed the measurement tools.

**Table 3 ijerph-19-16113-t003:** Correlations between the measured complexity and outcomes.

	Complexity Measurement Tools							
	INTERMED	INTERMED for Elderly	IM-E-SA	COMPRI	PCAM	COMPLEXedex	SWAAT	Bandini’s Tool
Patient outcomes	6 [19,20,21,22,23,24]	3 [25,26,27]	8 [27,29,30,31,32,33,34,35]	3 [20,21,36]	3 [37,38,40]	1 [41]	1 [42]	1 [43]
Health care costs		1 [25]	1 [28]					
Impacts on health care providers					1 [39]			

Note: 1–8 means the number of included studies. [] means reference number.

## Data Availability

The datasets generated during the current study are available from the corresponding author on reasonable request.

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
