# Peer review of "Evaluation of Complexity Measurement Tools for Correlations with Health-Related Outcomes, Health Care Costs and Impacts on Healthcare Providers: A Scoping Review"

_ijerph, 2022, doi:10.3390/ijerph192316113_

Round 1
Reviewer 1 Report
The project that is presented seeks to analyze the different evaluation tools of the complex patient in what refers to the correlations between patient complexity and outcomes, including health-related patient outcomes, healthcare costs, and impacts on healthcare providers.
It makes a good selection of articles, it is methodologically well designed and among its results the lack of studies that analyze the complexity of the patient from primary health care stands out, which is the natural health environment where the complex patient is cared for as a whole and from the social biopsychological context
Author Response
To Reviewer1,
Thank you for reviewing our manuscript.
Comment 1
Comments and Suggestions for Authors
The project that is presented seeks to analyze the different evaluation tools of the complex patient in what refers to the correlations between patient complexity and outcomes, including health-related patient outcomes, healthcare costs, and impacts on healthcare providers.
It makes a good selection of articles, it is methodologically well designed and among its results the lack of studies that analyze the complexity of the patient from primary health care stands out, which is the natural health environment where the complex patient is cared for as a whole and from the social biopsychological context.
Response 1
Many thanks for your supportive comments.
Reviewer 2 Report
This study identifies general patients' complexity measurement tools and points-out the studies which allowed the development of these tools, as well as studies which assessed the correlations between the suggested tools and outcomes.
In general, the framework and methodology are adequate and seem to have been meticulously applied. The literature review is extensive and well written. However, the readers would benefit from a more concise writing style. In particular, a shorter introduction and a slightly shorter Methods section will be appreciated.
Also, I would like to point-out some issues in the current version of this study:
1. The actual correlations between the various complexity measurement tools and outcomes were not included, and the authors state that further studies would be required in order to do so. However, I believe that the scientific value of this paper will significantly increase if its' results section will include a clear and simple statement of the strength, direction, and statistical significance of these correlations.
2. Line 75-76, the authors state that: "… primary care physicians, who are the most likely of healthcare workers to encounter complex patients". Can this statement be either defended or omitted?
3. Exclusion criteria (lines 115-119): while it is clear why end-of-life/palliative specific-tools were excluded, the choice to exclude other types of measures is unclear and should be justified (notable the exclusion of comorbidity/multimorbidity measurement tools).
4. Line 121: " the electric patient record systems in the United Kingdom.."- should be electronic
5. Lines 125-127: agreement (inter-examiner agreement) in measurement is referred to as "validity". However, it should be noted that these describe measures of reliability rather than validity.
Author Response
To Reviewer2,
Thank you for reviewing our manuscript.
Comment 1
This study identifies general patients' complexity measurement tools and points-out the studies which allowed the development of these tools, as well as studies which assessed the correlations between the suggested tools and outcomes.
In general, the framework and methodology are adequate and seem to have been meticulously applied. The literature review is extensive and well written. However, the readers would benefit from a more concise writing style. In particular, a shorter introduction and a slightly shorter Methods section will be appreciated.
Response 1
We understand the reviewer’s concern. We elaborated on "patient complexity" in the background because the concept would be unfamiliar to readers. In addition, we followed the JBI manual to describe the method. We believe that both of these descriptions are necessary. However, we do not feel too strongly about this and we are ready to shorten if the editor feels that the change would be better, and we would be glad to follow the editor’s decision. We also updated the introduction section to shorten each paragraph.
Comment 2
The actual correlations between the various complexity measurement tools and outcomes were not included, and the authors state that further studies would be required in order to do so. However, I believe that the scientific value of this paper will significantly increase if its' results section will include a clear and simple statement of the strength, direction, and statistical significance of these correlations.
Response 2
We appreciate the reviewer for giving us the opportunity to clarify this point. In the included studies, the relationship was reported in heterogeneous ways (e.g. p-value, r-squared, odds ratios, only the term “not significant”, etc.). We could not state these statistics in one ease way to understand the strength, or direction. Hence, we summarized in the present form of table 3 and Appendix B.
Comment 3
Line 75-76, the authors state that: "… primary care physicians, who are the most likely of healthcare workers to encounter complex patients". Can this statement be either defended or omitted?
Response 3
We decided to delete this sentence.
Comment 4
Exclusion criteria (lines 115-119): while it is clear why end-of-life/palliative specific-tools were excluded, the choice to exclude other types of measures is unclear and should be justified (notable the exclusion of comorbidity/multimorbidity measurement tools).
Response 4
Please see the response5 to the Editor’s Comment as follows:
In this review article, we defined patient complexity as “a typology of 4 overarching categories contributing to patient complexity: medical complexity, mental health disorders, socioeconomic factors, and individual patient behaviors or traits.” Therefore, we excluded some tools which mainly focused on biomedical aspects such as comorbidity/multimorbidity measurement tools, end-of-life/palliative specific measures. In addition, we intended to examine the patient complexity as "exposure". We excluded tools which focused on predicting frequent visit/hospitalization, and panel-weighting management tools machine learning prediction tools/algorithms, and software tools, because they are "outcomes".
(Before)2.2.2. Concept
We reviewed extant literature that evaluated patient outcomes (e.g., health out-comes, disease control, patient quality of life) and the impact on healthcare providers/systems (e.g., healthcare costs, administrative costs, healthcare professional burn-out) through complexity measurement tools. In order to cover related studies broader, we included not only observational studies but also interventional studies that could examine correlations between patient complexity stratified by tools and outcomes at baseline. While we included complexity measurement tools, we excluded the following measurement tools: comorbidity/multimorbidity measurement tools, frequent vis-it/hospitalization prediction tools, end-of-life/palliative specific measures, general condition panel-weighting management tools, machine learning prediction tools/algorithms, and software tools.
(After)2.2.2. Concept Line 136-145
We reviewed extant literature that evaluated patient outcomes (e.g., health outcomes, disease control, patient quality of life) and the impact on healthcare providers/systems (e.g., healthcare costs, administrative costs, healthcare professional burn-out) through complexity measurement tools. In order to cover related studies broader, we included not only observational studies but also interventional studies that could examine correlations between patient complexity stratified by tools and outcomes at baseline. We defined patient complexity as “a typology of 4 overarching categories contributing to patient complexity: medical complexity, mental health disorders, socioeconomic factors, and individual patient behaviors or traits.” Therefore, we excluded some tools which mainly focused on biomedical aspects such as comorbidity/multimorbidity measurement tools, end-of-life/palliative specific measures. In addition, we intended to examine the correlations between patient complexity as "exposure" and measured outcomes. We excluded tools which focused on predicting frequent visit/hospitalization, and panel-weighting management tools machine learning prediction tools/algorithms, and software tools, because they focused on predicting outcomes rather than measuring them or examining correlation with patient complexities.
Comment 5
Line 121: " the electric patient record systems in the United Kingdom.."- should be electronic
Response 5
Thank you for your pointing out this. We corrected it.
Comment 6
Lines 125-127: agreement (inter-examiner agreement) in measurement is referred to as "validity". However, it should be noted that these describe measures of reliability rather than validity.
Response 6
Thank you for your pointing out our misuse of terminology. We corrected it.
(Before) All validation studies other than those that addressed criterion-related validity, such as agreement in measurement results among examiners, were excluded because these validity studies did not contain outcomes such as patient health related outcomes, healthcare costs, or impacts on healthcare providers.
(After) Line 151-153 All reliability studies other than those that addressed criterion-related validity, such as agreement in measurement results among examiners, were excluded because these reliability studies did not contain outcomes such as patient health related outcomes, healthcare costs, or impacts on healthcare providers.
Reviewer 3 Report
The article under review presents the results of a scoping review of patient complexity measurement tools. It is well written and presented. The methodology used is appropriate, the protocol for identifying the publications on the subject is clearly set out and any digression necessary is mentioned. The results are useful for researchers interested in using such a tool to assess patient complexity. The discussion places the results in the context of patient management.
Three typing errors were identified: 1. CINHAL in the abstract should be CINAHL. 2. The word "electric" in page 3, line 121 should be electronic. 3. The sentence in page 13, lines 276-278 is repeated twice.
Recommendation: accept.
Author Response
To Reviewer3,
Thank you for reviewing our manuscript.
Comment 1
The article under review presents the results of a scoping review of patient complexity measurement tools. It is well written and presented. The methodology used is appropriate, the protocol for identifying the publications on the subject is clearly set out and any digression necessary is mentioned. The results are useful for researchers interested in using such a tool to assess patient complexity. The discussion places the results in the context of patient management.
Response 1
Many thanks for your supportive messages.
Comment 2
Three typing errors were identified: 1. CINHAL in the abstract should be CINAHL. 2. The word "electric" in page 3, line 121 should be electronic. 3. The sentence in page 13, lines 276-278 is repeated twice.
Response 2
We have corrected the errors as you kindly suggested.
Round 2
Reviewer 2 Report
Thank you for your reply. I have no further comments.
Author Response
To Reviewer2,
Comment 1
Thank you for your reply. I have no further comments.
Response 1
Many thanks for your review.